# Harmonizing Image Forgery Detection & Localization: Fusion of Complementary Approaches

**DOI:** 10.3390/jimaging10010004

**Published:** 2023-12-25

**Authors:** Hannes Mareen, Louis De Neve, Peter Lambert, Glenn Van Wallendael

**Affiliations:** Internet Technology and Data Science Lab (IDLab), Ghent University—imec, 9052 Ghent, Belgiumpeter.lambert@ugent.be (P.L.); glenn.vanwallendael@ugent.be (G.V.W.)

**Keywords:** multimedia forensics, image forgery detection, image forgery localization, fusion, complementariness

## Abstract

Image manipulation is easier than ever, often facilitated using accessible AI-based tools. This poses significant risks when used to disseminate disinformation, false evidence, or fraud, which highlights the need for image forgery detection and localization methods to combat this issue. While some recent detection methods demonstrate good performance, there is still a significant gap to be closed to consistently and accurately detect image manipulations in the wild. This paper aims to enhance forgery detection and localization by combining existing detection methods that complement each other. First, we analyze these methods’ complementarity, with an objective measurement of complementariness, and calculation of a target performance value using a theoretical oracle fusion. Then, we propose a novel fusion method that combines the existing methods’ outputs. The proposed fusion method is trained using a Generative Adversarial Network architecture. Our experiments demonstrate improved detection and localization performance on a variety of datasets. Although our fusion method is hindered by a lack of generalization, this is a common problem in supervised learning, and hence a motivation for future work. In conclusion, this work deepens our understanding of forgery detection methods’ complementariness and how to harmonize them. As such, we contribute to better protection against image manipulations and the battle against disinformation.

## 1. Introduction

Manipulating images is easier than ever using tools such as Adobe Photoshop or recently accessible AI-based tools. This has raised significant concerns about the spread of deceptive content, which poses risks when used for misinformation, as false evidence, or for other fraudulent purposes. The potential misuse of manipulated images underscores the pressing need for robust image forgery detection methods. These methods serve as a crucial line of defense in safeguarding the integrity of visual information and mitigating the social ramifications of image-based deception.

While substantial progress has been made in the development of image forgery detection techniques [1,2], a considerable gap remains in their capacity to effectively detect forgeries in real-world scenarios. That is, manipulated images found in the wild are often subjected to alterations that were not seen in a detection method’s training set. For example, methods targeting deepfake (face) forgery detection only [3,4,5] may not be able to detect other types of fakes, such as splicing or copy-move attacks. In other words, generalization is a major challenge of image forgery detection methods. Additionally, the detection performance is hindered by post-manipulation processing, such as decreasing the quality due to compression when shared on social media. Robustness against such (intentional or unintentional) attacks is another major challenge of forgery detection techniques.

Each individual detection method has its own target use case, strengths and limitations. They often operate under specific assumptions and are trained on a certain manipulation trace or forgery type. As a result, forensic investigators have to run and interpret multiple detection methods, which increases their required skill level and makes the investigation harder. Because of the complementary assumptions and methodologies of existing methods, one can expect a boost in performance when combining multiple methods to maximize the number of utilized forensic clues [1]. Additionally, interpreting a single output requires less skill than interpreting multiple outputs.

Existing fusion methods that combine image forgery detection methods typically have some limitations [6,7,8,9,10,11,12]. First, they may not provide generic frameworks to easily add existing forgery detection methods. This would be particularly interesting when new high-performing methods are developed later, and may, therefore, greatly increase the overall performance of the fusion model. However, some existing fusion methods perform a unique post-processing workflow on each individual method [8], adapt the individual method [7], or target the fusion method on different versions of the same methodology or architecture [6,11,12]. This hurdles the later addition of new individual methods to the fusion framework. Second, existing fusion methods often do not include recent deep-learning based methods that demonstrated better performance than the conventional methods that are used instead. Third, existing fusion methods select complementary methods based on their complementary forgery traces. However, they may lack an objective measurement of their complementariness on evaluation datasets. These limitations of existing fusion methods are further discussed in Section 2.2.

This paper explores the synergy of complementary image forgery detection methods and proposes a novel fusion approach to enhance their performance. The main contributions are the following. First, we objectively measure the complementariness of existing forgery detection and localization methods on several evaluation datasets. Second, we propose a novel fusion method based on a deep-learning architecture that combines the heatmaps of existing forgery localization methods. We take special care to provide a generic fusion approach in which new detection methods can be added in a later stage, without the need to develop a unique post-processing workflow for them. Additionally, we include both conventional and more recent high-performing data-driven detection methods. As such, we tackle the limitations of existing fusion methods.

The rest of this paper is organized as follows. First, Section 2 discusses related image forgery detection and fusion methods. Then, we measure and analyze the complementariness of existing forgery detection methods in Section 3. Subsequently, Section 4 proposes a novel fusion method. The performance of the proposed fusion method is experimentally evaluated in Section 5. Finally, the paper is concluded in Section 6.

## 2. Related Work

This section first provides an overview of the classes of existing image forgery detection and localization algorithms in Section 2.1, with a specific focus on the algorithms utilized in the proposed fusion method. Additionally, existing fusion methods are discussed in Section 2.2.

### 2.1. Image Forgery Detection and Localization Methods

Image forgery detection methods can be classified based on the type of forensic clue or artifact that they exploit. The two main forensic types are acquisition artifacts and compression artifacts.

Acquisition artifacts originate from the image capture process. Methods exploiting these artifacts focus on anomalies introduced during the transformation of a physical scene into a digital image. Camera lenses, for instance, possess inherent imperfections. These imperfections are harnessed for forgery detection [13,14]. Furthermore, the demosaicing process, which interpolates data from Color Filter Arrays (CFAs), introduces correlation patterns into the resulting image pixels. Any manipulation can distort this correlation pattern, thereby providing a potential signature of tampering [15]. Moreover, methods exploiting photo-response non-uniformity (PRNU) noise have been developed to identify both camera models and manipulations by analyzing inconsistencies in the unique patterns left by the camera sensor [16]. While exploiting the PRNU noise is effective, it requires multiple images from the same camera, which may not always be feasible in practical scenarios. Noiseprint [17] overcomes this limitation by extracting a camera model fingerprint without extensive data from a specific camera. That is to say, during training, Noiseprint used a large dataset of cameras to learn a camera model fingerprint, but during inference, one image was sufficient. Noiseprint is a machine learning algorithm trained using a Siamese network, and the training set contains only real data, hence there is no bias towards specific forgery types that were seen during training. This is what one calls one-class learning. As such, Noiseprint is a practical and generalizable solution. Similar to Noiseprint, SpliceRadar [18] also uses camera model features in a one-class learning architecture. Moreover, EXIF-SC [19] also uses one-class learning, and aims to determine whether every part of the image content is self-consistent with its EXIF metadata.

A second common approach in image forgery detection involves examining anomalies in JPEG compression artifacts. JPEG is a widely used image compression standard. Mismatches in JPEG blocks and the corresponding grid are signs of manipulation [20,21,22]. Conventional detection methods have relied on statistical analysis of these coding artifacts, leading to the creation of handcrafted models. However, the applicability of these models in real-world scenarios remains a challenge due to their assumptions and limitations. An alternative approach involves double quantization or double JPEG compression artifact-based methods [23,24,25,26]. These methods assume that the authentic region is compressed twice, while the manipulated region is compressed only once. This concept has been explored using both conventional methods and deep-learning approaches, where neural networks are trained to automatically extract relevant features. For example, Comprint extracts a compression fingerprint using deep learning, in which inconsistencies suggest manipulation [11]. This is similar to Noiseprint [17], with the difference that it uses compression traces rather than acquisition artifacts. More recent advancements, such as the Compression Artifact Tracing Network (CAT-Net), have employed deep learning and large training sets to exploit both acquisition and JPEG compression artifacts, operating in both the spatial and Discrete Cosine Transform (DCT) domain, offering improved detection capabilities [27]. However, finding sufficient training datasets that cover and generalize to real-world circumstances is a main unsolved challenge in image forgery detection and localization.

In general, deep-learning-based methods have demonstrated high performance. However, there is still a significant gap to close to reach 100% accuracy, especially under in-the-wild circumstances.

### 2.2. Fusion Methods

A key challenge of fusion is identifying algorithms that complement each other, leading to improved overall robustness and reliability of localization and detection. Several fusion approaches have been proposed in the literature, and they can be categorized as either feature-level or pixel-level fusion.

#### 2.2.1. Feature-Level Fusion

Feature-level fusion occurs at the same level where image forgery detection and localization algorithms extract suitable low-level features from the image. An example of feature-level fusion is Comprint + Noiseprint [11]. Comprint and Noiseprint are complementary strategies, with Comprint exploiting compression artifacts and Noiseprint targeting camera-based acquisition artifacts. To fuse the methods, their localization algorithm is adapted by concatenating the features (i.e., fingerprints) of both Comprint and Noiseprint, and applying expectation-maximization on the concatenated feature array. Comprint + Noiseprint outperforms the individual Comprint and Noiseprint methods. Although feature-level fusion may perform well, such approaches often encounter challenges related to feature selection and management of a large number of features when combining a significant amount of algorithms [1]. Additionally, it is not straightforward to add other existing methods, as this may require transforming those methods into features in the same latent space. As such, pixel-level fusion methods may provide an easier framework for fusion.

#### 2.2.2. Pixel-Level Fusion

In pixel-level fusion, the output heatmaps of the individual forgery localization methods are combined. For example, Korus and Huang proposed to run forgery localization at multiple scales and combine those results [6]. Similarly, Li et al. fused a statistical feature-based approach with a complementary copy-move detection by combining the corresponding tampering possibility maps [7]. Additionally, Liu and Pun proposed a deep fusion network in which two complementary base networks were trained, and subsequently combined in a fusion network [12]. The base networks are complementary because they are each trained on different forgery traces (noise and compression). These methods have in common that it is not straightforward to add other existing methods, since they adapt the individual methods, or target the fusion method on different versions of the same methodology or architecture.

Iakovidou et al. proposed a knowledge-based fusion approach that combines several selected conventional detection algorithms [8]. These algorithms were selected based on their performance and based on the complementarity of their assumptions. This fusion method first refines the individual localization heatmaps by normalizing, binarizing, connecting, and filtering them. Then, statistical features are extracted from these refined maps to automate the evaluation of their usefulness in a fusion unit. The fusion takes into account estimates of the methods’ interpretability, inter-compatibility, reliability (based on previous experiments), and confidence. The fusion approach primarily focused on conventional detection algorithms, while better-performing data-driven algorithms were not evaluated. Additionally, this fusion approach incorporates certain hard-coded heuristics and assumptions about the selected input algorithms, which limits its practical adaptability to adding new forgery detection algorithms.

Some recent fusion methods perform supervised deep-learning directly on the heatmaps (i.e., the individual methods’ outputs). For example, Operation-wise Attention Fusion (OwAF) [9] improves upon the work of Iakovidou et al. [8], by combining conventional methods in a simple CNN architecture for supervised learning. In numerous cases, the fusion outperforms the individual algorithms, as well as the older fusion algorithm [8]. Similarly, Siopi et al. proposed a multi-stream fusion network that combines two existing complementary conventional detection algorithms (one based on noise and one based on compression) [10]. The input image and the output heatmaps of the selected methods are processed by different encoder stream networks, and the encoding outputs are subsequently fused. These fusion methods make it more practical to add new existing detection methods to the set of input methods. However, a limitation of these models is their lack of generalization to manipulations unseen during training, which is a common problem in supervised learning [28]. Consequently, it occasionally yields worse results compared to the best individual input detection algorithm. Furthermore, the absence of data-driven algorithms in the fusion is a recurring observation among existing fusion methods, despite their demonstrated superior performance over conventional detection algorithms.

In summary, the current state-of-the-art does not offer a universal solution that addresses all challenges in image forgery detection and localization. Existing fusion algorithms exhibit several limitations, as they often rely on predefined heuristics and do not cover recent data-driven algorithms. Additionally, the complementariness of the individual methods is never objectively measured but is only motivated based on the assumptions that they make. Hence, further advancements in this field are needed to develop more robust and effective fusion approaches.

## 3. Selection & Complementariness of Image Forgery Detection Methods

This section discusses the selected image forgery detection methods and measures their complementariness. Note, that we objectively measure the complementarity, which is novel compared to previous fusion methods that typically select complementary methods solely based on the complementary assumptions that the individual methods make (i.e., they are based on complementary forensic clues).

This section is organized as follows. First, the experimental setup of this analysis is described in Section 3.1. Then, the results of this analysis are reported and discussed in Section 3.2.

### 3.1. Experimental Setup

#### 3.1.1. Image Forgery Detection Algorithms

In this study, we evaluate a subset of conventional detection algorithms (ADQ1, DCT, BLK, CAGI) that were also chosen in state-of-the-art fusion methods [8,9]. Additionally, we add more recent, high-performing, data-driven algorithms (Noiseprint, Comprint, Comprint + Noiseprint, and CAT-Net). It is noteworthy that Comprint + Noiseprint in itself is already a feature-level fusion method. All selected algorithms are listed in Table 1. Some algorithms are based on JPEG compression artifacts, whereas others on camera-based acquisition artifacts.

#### 3.1.2. Evaluation Datasets

We selected a diverse set of datasets containing manipulated images, on which we evaluated the (complementary) performance of the selected forgery detection methods. These datasets cover a wide range of manipulations, created by both traditional editing software and artificial intelligence (AI) techniques. In total, we selected seven datasets, listed in Table 2, each presenting different challenges and characteristics. Additionally, some subsets of those datasets are used for training (see Section 5.1.1).

#### 3.1.3. Performance Metrics

To evaluate the performance of the image forgery localization methods on the evaluation datasets, we measure the pixel-level forgery localization performance using the F1 score. The F1 score is the harmonic mean of the precision and recall, and is calculated as follows:(1)$$F1=\frac{1}{\frac{1}{\mathrm{precision}}+\frac{1}{\mathrm{recall}}}=\frac{2\mathrm{TP}}{2\mathrm{TP}+\mathrm{FN}+\mathrm{FP}}$$

In Equation (Equation 1), TP, FP and FN represent the number of pixel predictions that are True Positives, False Positives, and False Negatives, respectively. Note, that a higher F1 score (i.e., closer to 1) is better than a lower one (i.e., closer to 0). To classify pixels as either real or fake, the corresponding value in the heatmap needs to be compared to a certain threshold. To eliminate the threshold’s influence on the calculation, the maximum F1 score across all possible thresholds is reported in this paper. Additionally, it is essential to account for cases where the heatmap’s polarity is inverted relative to the ground truth. To address this, we consider both the original and inverted heat maps, and report the maximum F1 score.

Note, that in this section, we only measure the pixel-level localization performance, and we do not (yet) consider the image-level detection performance. When evaluating the proposed fusion method in Section 5, the image-level detection performance metric is described and additionally evaluated.

#### 3.1.4. Theoretical Oracle Fusion & Complementary F1 Delta Metric

To evaluate how complementary a set of forgery localization methods are, we define two new concepts: the theoretical oracle fusion, and the Complementary F1 Delta metric.

The theoretical oracle fusion is the maximum F1 score of all methods, taken for each image separately. In essence, it is an oracle that, for each image in a certain dataset, knows which is the best-performing forgery localization method, and selects that one as its output. In other words, a different algorithm may be selected for each image. As such, the oracle fusion can be seen as a target score of a fusion algorithm on a certain dataset.

The Complementary F1 Delta is a measure of complementariness between two detection methods. It calculates the improvement in F1 score between using one of the two methods as a reference algorithm, on the one hand, and the theoretical oracle fusion of both methods, on the other. When the Complementary F1 Delta is zero, it means that the reference method performs equal or better than the second method for every image in the dataset. A larger Complementary F1 Delta signifies that the two detection methods are more complementary, and hence each shows strengths in different scenarios.

### 3.2. Performance & Complementariness Results

We measured the pixel-level forgery localization performance (F1 score) of each forgery detection method on each evaluation dataset. Additionally, we measured the Complementary F1 Delta for each combination of two methods. These results are shown in Appendix A, i.e., in Table A1, Table A2, Table A3, Table A4, Table A5, Table A6 and Table A7 for the VIPP, IMD2020, DSO-1, OpenForenics, FaceSwap, Coverage, and NC2016 dataset, respectively. These tables highlight the three largest Complimentary F1 Delta values for each row in **bold**, underline and *italics*, respectively. The same highlighting is used for the column with F1 scores.

From the tables in Appendix A, we notice that the recent CAT-Net method performs best on all datasets, except for the DSO-1 dataset where Comprint + Noiseprint performs best. This showcases that these more recent data-driven detection methods perform better than conventional detection methods. Recall that existing fusion methods often limit themselves to combining existing conventional detection methods. Hence, including these newer, high-performing data-driven methods is one of the contributions of our work.

Regarding the complementariness of CAT-Net, we can see the Complementary F1 Delta when combining CAT-Net with each of the other methods in the first row of each table in Appendix A. On all datasets, combining CAT-Net with Comprint + Noiseprint results in the largest localization performance improvement (except for OpenForensics, where combining CAT-Net with Comprint performs slightly better). For example, on the FaceSwap dataset in Table A5, the individual F1 score of CAT-Net is 0.449, and the individual F1 score of Comprint + Noiseprint is 0.412. By combining these two methods using the theoretical oracle fusion, we can increase CAT-Net’s F1 score by 0.130. This demonstrates the complementarity of these two high-performing methods.

Although the tables in Appendix A showcase the complementarity of each *two* methods, they do not consider the fusion of *all* methods. Therefore, we additionally show the localization performance of theoretical oracle fusion of all methods, in Table 3. This table shows the F1 scores of each individual method (again), in addition to the best performance of fusing just *two* methods, and the performance of fusing *all* methods, using the theoretical oracle. Moreover, the last row of the table shows the performance of the proposed fusion method, which is irrelevant in this section, and is further discussed in Section 5.2.1. Note, that in this table, we highlight the three largest values for each column in **bold**, underline and *italics*, respectively. It is no surprise that the first and second largest values of each column are the oracle fusion rows, as these take the image-wise maximum of the individual methods.

From Table 3, we notice that the F1 score can be improved by combining two methods, and even more so when combining all selected methods. For example, CAT-Net has the best individual localization performance on the NC2016 dataset (F1 score of 0.487). The best oracle fusion of 2 methods increases this F1 score to 0.594 (by combining CAT-Net with Comprint + Noiseprint, see Table A7). Moreover, the oracle fusion of *all* methods brings the performance to an F1 score of 0.626. This is a significant boost in performance, demonstrating the potential of fusing all methods. Hence, in Section 4, we propose a practical fusion method that combines all these detection methods.

In summary, this section demonstrated the potential of fusing existing image forgery detection and localization methods. It should be noted that we analyzed the performance using a theoretical oracle fusion, which cannot be applied in real-world scenarios. Instead, it serves only as a demonstration of fusion potential and aids in measuring the complementariness of existing methods. As such, we motivate combining the selected methods in our proposed fusion method, in Section 4.

## 4. Proposed Fusion Method

In this section, we propose a novel pixel-level fusion method that combines the outputs of the selected image forgery detection and localization methods listed in Table 1, in addition to the investigated image.

We interpret the problem of fusing multiple heatmaps into a single heatmap as an image-to-image translation task. A widely adopted deep-learning architecture for addressing image-to-image challenges is the Generative Adversarial Network (GAN) [36,37,38,39,40]. In our context, a conditional Generative Adversarial Network (cGAN) serves as the foundation of the proposed fusion algorithm. A GAN consists of two fundamental models: a Generator (G) and a Discriminator (D). These two models engage in a competitive dynamic, with the generator striving to generate the fused heatmaps as close as possible to the ground truth, while the discriminator aims to distinguish the difference between the ground truth and fused heatmaps. In the context of GANs, the goal is to learn a transformation from a random noise vector *z* to an output image *y*, denoted as G:z→y. However, given our objective of using heatmaps as input, the utilization of a conditional GAN (cGAN) is more fitting. This type of GAN utilizes additional conditional information, *x*, resulting in G:{x,z}→y, in which *x* is the concatenation of the heatmaps and original image [41].

In the following sections, we delve into the employed training methodologies and architectures (Section 4.1). Subsequently, we discuss how the fusion algorithm is used during inference (Section 4.2).

### 4.1. Training Methodology

Our proposed fusion method is based on a similar training objective as typical *cGANs*, such as the Pix2Pix method. Pix2Pix is a widely used model in image-to-image translation [42]. In general, the main objective of a *cGAN* is to identify the optimal generator ($G^{*}$) capable of generating outputs that closely resemble the distribution of real data (i.e., the ground truth forgery masks). In our proposed method, finding the optimal generator $G^{*}$ is formulated as follows:(2)$$G^{*}=\arg\min_{G}\max_{D}L_{cGAN}\left(G,D\right)+\lambda L_{L2}\left(G\right).$$

In other words, *G* seeks to minimize the *cGAN* loss against the adversarial *D*, which in contrast strives to maximize the loss.

In Equation (Equation 2), we combine the typical loss function of a cGAN $L_{cGAN}$ with the L2 loss $L_{L2}\left(G\right)$. These loss functions are expressed as follows:(3)$$L_{cGAN}=E_{x,y}\left[\log D(x,y)\right]+E_{x,z}\left[\log\left(1-D(x,G(x,z)\right)\right],$$
(4)$$L_{L2}\left(G\right)=E_{x,y,z}[||y-{G\left(x,z\right)||}_{2}].$$

By including the *L*2 loss, the generator has an additional goal of producing images that closely resemble the ground truth. Note, that we utilize the *L*2 loss, which is in contrast to Pix2Pix which uses the *L*1 loss [42]. In Pix2Pix, the *L*1 loss was chosen to minimize image blurring. However, blurriness has less impact on performance in our forgery localization task. Instead, using the *L*2 loss intensifies the penalty for generating noisy heatmaps, amplifying regions with significant errors and disregarding parts with negligible error, due to the quadratic nature of the *L*2 distance.

It is additionally worth noting that the actual input no longer involves random noise (*z*), as this approach was found to be ineffective [42]. Instead, noise is introduced through dropout, applied across multiple layers of the generator [42]. This is conducted both during training and inference (the latter is further discussed in Section 4.2).

The generator and discriminator are individually discussed in depth in Section 4.1.1 and Section 4.1.2, respectively.

#### 4.1.1. Generator

The high-level process of training the generator is visualized in Figure 1. The generator loss consists of two parts.

First, it aims to minimize the difference between the generator’s output (fused heatmap) and the *discriminator’s expectations*. This is achieved by using a sigmoid-cross-entropy loss of the discriminator’s output and an array of ones (which signifies the discriminator’s expectation for every patch to originate from the ground truth). A low score indicates effective deception of the discriminator and vice versa.

Second, the generator aims to minimize the difference between the fused heatmap and the *ground truth*. This is achieved by using the L2 loss, also known as the Mean Squared Error (MSE), between the fused heatmap and the ground truth. The overall loss is the sum of the sigmoid-cross-entropy loss and the MSE (the latter first multiplied by lambda, which is set to λ=100, as conducted in Pix2Pix [42]). This combined loss adapts the generator’s neural network weights during training, causing it to become better at generating good fused heatmaps.

The generator’s architecture is based on U-Net [43], with some modifications. The U-Net architecture involves two main parts: an encoder which iteratively downsamples the previous layers, and a decoder which iteratively upsamples the layers back to the original resolution. Each iterative component contains convolution, batch normalization and Rectified Linear Unit (ReLu) layers. Moreover, the first three upsample components include a dropout layer in between the batch normalization and ReLu. This dropout accounts for randomness, as discussed in Section 4.1. Additionally, skip connections link the corresponding decoder and encoder blocks at the same resolutions. These connections concatenate their channels, allowing information to be shared between them.

#### 4.1.2. Discriminator

The high-level process of training the discriminator is visualized in Figure 2. The discriminator’s loss is separated into two parts.

First, the *fused heatmap* is presented to the discriminator along with the input images (i.e., the heatmaps and investigated image). The discriminator’s output is then employed to calculate the sigmoid-cross-entropy loss against an array of zeros (which, in contrast to an array of ones, signifies the discriminator’s expectation for every patch to be originating from the fused heatmap, and not the ground truth). This constitutes the *generated loss*.

Second, the *ground truth* and the input images are jointly supplied to the discriminator. The discriminator’s output is again utilized to compute the sigmoid-cross-entropy loss, yet this time against an array of ones (signifying the expectation to be originating from the ground truth). This constitutes the *real loss*. The final discriminator loss is the sum of the generated loss and the real loss.

The discriminator’s architecture is based on a convolutional PatchGAN classifier [42], in order to classify whether image patches originate from either a fused heatmap or the ground truth. This architecture consists of multiple downsampling blocks responsible for creating patches. These downsampling blocks also follow the convolution, batch normalization, and ReLu structure. Additionally, a layer of spectral normalization [44] is introduced after each convolutional layer. This spectral normalization layer re-normalizes the weights of the convolutional layer whenever they are updated, creating a network that mitigates gradient explosion problems, and decreasing the discriminator’s rapid learning rate. Consequently, it allows the generator more time to learn during the initial epochs, where it may struggle.

### 4.2. Inference Methodology

When applying the model in practice, on real-world images, the fusion model runs in inference mode. The inference process of our fusion algorithm mirrors that of the Pix2Pix architecture. Specifically, only the generator component of the algorithm is utilized.

Note, that our proposed model utilizes the dropout layers of the generator during its inference. This is in contrast to a typical inference protocol, in which dropout layers are not utilized during inference, but only during training. However, in the context of the random nature inherent to (c)GANs, the dropout layer continues to be employed during inference. This was also conducted in Pix2Pix [42].

As such, the fusion algorithm can effectively process new images and produce fused heatmaps that are well-suited to real-world scenarios.

## 5. Evaluation of Proposed Fusion Method

This section evaluates the proposed fusion method described in Section 4. First, the experimental setup is described in Section 5.1. Then, the results of forgery localization and forgery detection performance are discussed in Section 5.2.

### 5.1. Experimental Setup

#### 5.1.1. Training Setup

For this study, the OpenForensics [30], FaceSwap [31], and IMD2020 [29] datasets of manipulated images are used for training the model. These datasets were chosen due to their size and wide coverage of diverse forgery types. This leaves the other datasets listed in Table 2 purely for evaluation purposes. In the selected datasets for training, a train/validation/test split is made, distributing the data with an 80/10/10 ratio. Images are randomly selected from the datasets for each split. Additionally, to create a balanced dataset, we included an equal number of training and validation images from each of the datasets. Among the three, FaceSwap is the smallest dataset, having 703 training and 88 validation images. As such, the same quantities of OpenForensics and IMD2020 data are used during training, resulting in a training dataset of 2109 images and a validation dataset of 264 images. Note, that the test datasets are not artificially limited, and remain 10% of the size of the full datasets.

For every image in the dataset, both the image itself and all heat maps produced by the selected input algorithms from Table 1 are utilized as input. It is noteworthy that one of the input algorithms is Comprint + Noiseprint, which is a feature-level fusion method. As such, we combine feature-level and pixel-level fusion. Before using these input images for training, a preprocessing stage is conducted, similar to in Pix2Pix [42]. That is, all heatmaps and the investigated image are first resized to a resolution of 256 × 256 pixels. Afterward, the training set is augmented by resizing the images to 286 × 286 and then randomly cropping them back to the size of 256 × 256. Additionally, random mirroring is implemented by horizontally flipping the image (left to right). All inputs are concatenated at the channel level, generating a composite input size of 256 × 256 × 11 pixels for each image. Eleven channels are the result of concatenating the three RGB channels of the input image with the 8 channels of the single-channel heatmaps.

The actual training exists out of 40 epochs with a batch size of 1, yielding 2109 steps per epoch. We adopt the Adam solver, following the parameters used in Pix2Pix [42]: a learning rate of 0.0002, and momentum parameters $\beta_{1}=0.5$ and $\beta_{2}=0.999$.

#### 5.1.2. Evaluation Setup

The datasets used for evaluating the model’s performance are the same as those discussed in Section 3.1.2, and listed in Table 2. However, the datasets used for the training of the model have been swapped for the newly-created test split. The other evaluation datasets remain unchanged.

In addition to the pixel-level forgery localization performance metric (F1 score), we additionally measure the image-level forgery detection performance to evaluate the fusion model. As an image-level forgery detection measure, we utilize the Area Under the ROC Curve (AUC). That is, first, a global statistic is extracted from the heatmap. Next, this global statistic is compared to a threshold, which determines whether the algorithm predicts the image to be forged or pristine. To make the evaluation independent of a certain threshold, the Receiver Operating Characteristic (ROC) curve is used, which illustrates the TP rate against the FP rate at different threshold values for each dataset. Afterward, the ROC is used to compute the AUC, which integrates the ROC curve over all threshold values, yielding a single value from 0 to 1. An AUC of 0.5 indicates random performance. Higher AUC values indicate better detection capabilities. To ensure the evaluation’s independence from the choice of the global statistic, the global AUC score is computed as the maximum AUC among several considered global statistic options, namely the average, median, standard deviation, 99.5th percentile, and 97.5th percentile. This approach mitigates any potential bias that might arise from favoring a particular global statistic.

In addition to comparing the performance of the proposed fusion method to the individual methods’ performance, and the theoretical oracle fusion performance, we also compare it to two simple fusion strategies: *Maximum* and *Minimum*. These fusion methods transform the input heatmaps from the image forgery detection methods into a fused heatmap. Each pixel value of a fused heatmap is the maximum (or minimum, respectively) of the corresponding pixel values in the input heatmaps. For these simple fusion methods, we selected only CAT-Net and Comprint + Noiseprint, which are the two best performing forgery detection methods (as discussed in Section 3.2. We do this because we noticed noisy fused heatmaps with worse performance when additionally using the other 6 input methods.

Note, that we do not compare the results of our proposed method to related fusion methods. That is because existing methods fuse other methods than in our work, often only focusing on conventional methods. Moreover, in many state-of-the-art fusion methods, it is not straightforward to add new detection methods as input. Even when it is relatively more easy, it may require re-training those models. Finally, existing fusion methods are typically not available open-source.

### 5.2. Results

We discuss the performance of forgery localization and detection in Section 5.2.1 and Section 5.2.2, respectively.

#### 5.2.1. Image Forgery Localization Performance

The F1 scores for forgery localization of the proposed fusion method are given in the last row of Table 3. Recall that, in this table, we highlight the three largest values for each column in **bold**, underline and *italics*, respectively. Additionally, recall that the oracle fusion models are theoretical and cannot be applied in practice. Instead, they only serve as a theoretical target that demonstrates the fusion potential. The proposed fusion method is not evaluated on the full IMD2020, OpenForensics and FaceSwap datasets, as it was trained on a portion of these datasets. Instead, the F1 score of the *test splits* of IMD2020, OpenForensics and FaceSwap are given in Table 4.

The proposed fusion algorithm demonstrates exceptional localization performance on the test splits of the IMD2020, OpenForensics, and Faceswap datasets in Table 4. Impressively, it even surpasses the target of theoretical oracle fusion of all methods. Additionally, it also surpasses the Maximum and Minimum fusion methods.

In contrast to the test splits of the datasets that were used during training, evaluating the localization performance on previously unseen datasets performs less impressively, as presented in Table 3. On those datasets, the proposed fusion method outperforms the individual forgery localization methods for the VIPP and NC2016 dataset, but scores worse than the best individual method for the DSO-1 and Coverage dataset. Additionally, the oracle fusion methods scored higher than the proposed fusion methods in all those datasets. This demonstrates the lack of generalization on unseen datasets, which is also an issue in related work that uses supervised learning [9,10].

In our case, the generalization issue may be specifically attributed to the substantial performance gap between CAT-Net and the other input algorithms, which is especially prevalent on the IMD2020 and OpenForensics training datasets, while less outspoken on the FaceSwap training dataset. Consequently, this creates a bias towards the heatmaps generated by the CAT-Net algorithm. This lack of generalization is also an implication of using two-class learning in image forgery localization/detection, in contrast to one-class learning where no fake images are seen during training [1,11,17]. In such two-class learning methods, obtaining a training dataset that contains all possible forgeries is impractical. Although the method does not generalize very well to unseen datasets, it should be noted that the performance of the proposed fusion method is still relatively good. Most notably, the proposed method performs better than all individual input methods on two datasets (VIPP and NC2016). On the other datasets (DSO-1 and Coverage), the proposed fusion method still obtains an F1 score between the top-performing methods, and may hence still help to provide forensic investigators with a single reliable heatmap, rather than a large set of heatmaps from all individual methods. As such, the proposed fusion method can be used in practical applications.

When comparing the proposed fusion method to the simple Maximum and Minimum fusion methods, we make the following observations. For the DSO-1 and NC2016 methods, the simple fusion methods showcase a slightly better performance. However, for the VIPP and Coverage datasets, as well as the test splits of IMD2020, OpenForensics, and Faceswap, the proposed fusion method outperforms the simple fusion methods.

Visual examples of the forgery localization heatmaps can be observed in Figure 3. These examples demonstrate the ability of the fusion method to successfully learn to combine the heatmaps of the individual methods into a more accurate fused heatmap.

#### 5.2.2. Image Forgery Detection Performance

The AUC scores for forgery detection are given in Table 5. Note, that the OpenForensics dataset is not present in the table, as it does not contain real images and can, therefore, not be evaluated on the binary forgery detection task. Additionally, note that we evaluate the *full* datasets (as in Table 3), and not the *test splits* of the datasets (as in Table 4).

In Table 5, we notice similar trends as seen in the forgery localization performance discussion in Section 5.2.1. That is, the proposed fusion method obtains relatively high AUC values, but does not perform better than the best individual method. Notably, on DSO-1, a significant gap is observed between the best individual method and the proposed fusion method. This worse performance can again be attributed to the fusion method being biased towards CAT-Net heatmaps, as also discussed in Section 5.2.1. That is, on DSO-1, CAT-Net’s performance is particularly worse than Comprint, Noiseprint, and Comprint + Noiseprint, whereas it is typically the opposite in the other evaluation datasets and in the training datasets. As such, when CAT-Net performs badly, the bias towards CAT-Net has a particularly negative effect on the observed fusion detection performance. Despite the fusion method not performing better than the best individual method, it is still a relatively well-performing method and is positioned in second place for the VIPP, Coverage, and NC2016 datasets. Moreover, in comparison with the simple fusion methods (Maximum and Minimum), the proposed fusion method showcases worse AUC values for the DSO-1 and NC2016 datasets, but better results for the VIPP and Coverage datasets. This is the same observation we made when discussing the image forgery localization performance in Section 5.2.1.

In general, this section demonstrated that the proposed fusion method is a viable approach to combine the outputs of multiple individual methods into a single fused heatmap.

## 6. Conclusions

This study aims to enhance image forgery detection and localization methods, by uniting existing forgery detection methods.

We evaluated the performance and complementariness of selected existing methods, focusing on both conventional and more recent data-driven methods. We objectively demonstrated the potential to combine multiple detection methods on the basis of the theoretical oracle performance and Complementary F1 Delta metric, which are novel concepts proposed for the first time in this paper.

Moreover, we propose a deep-learning-based fusion method that is trained using cGAN architecture. Our approach shows good results in enhancing forgery localization. Although it sometimes struggles to generalize on unseen datasets, its performance is relatively high and can help forensic investigators with the interpretation of the results of forgery detection and localization methods.

Future work may tackle the generalization issue by utilizing more training data, from a larger variety of datasets with a larger variety of forgery types. Additionally, the generator architecture in our proposed framework could be replaced with cutting-edge alternatives such as vision transformers [45] in order to further refine the performance, which can be inspired by related work that evaluated the generalization of deep-learning models [28]. Finally, we can improve the overall fusion performance by including new high-performing methods, such as TruFor [46] and FOCAL [47], which are relatively easy to include in our proposed fusion framework.

In conclusion, this work contributes to a deeper understanding of image forgery detection and localization. We showcased their complimentariness, as well as both the promising results and limitations of fusion. As such, we further protect our society from misinformation, false evidence, and fraud.

## Figures and Tables

**Figure 1 jimaging-10-00004-f001:**
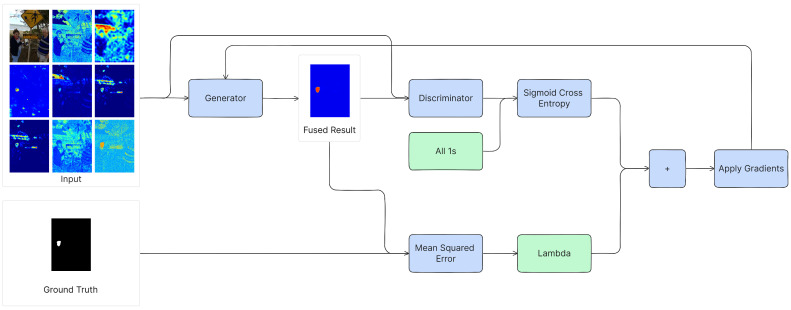
The training procedure of the generator.

**Figure 2 jimaging-10-00004-f002:**
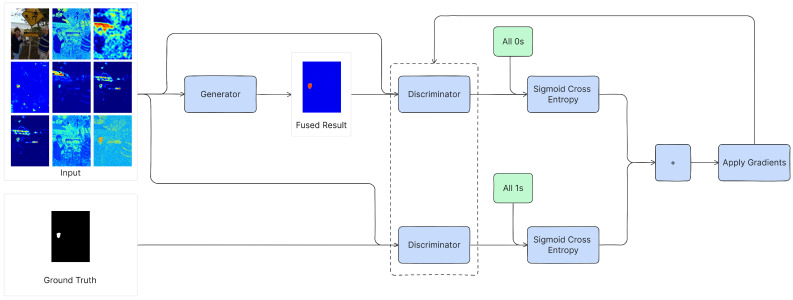
The training procedure of the discriminator. The dotted box around the discriminator signifies that the weights of these discriminators are shared.

**Figure 3 jimaging-10-00004-f003:**
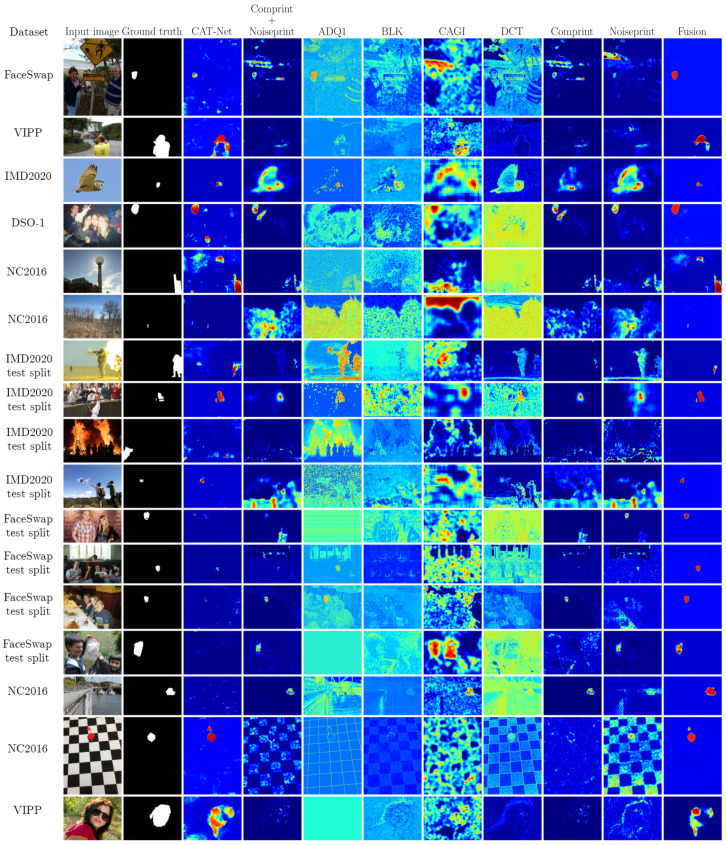
Examples of forged images and the corresponding heatmaps by the input forgery localization algorithms and the proposed fusion method.

**Table 1 jimaging-10-00004-t001:** Selection of input algorithms that are evaluated on their performance and complementarity, and used in the proposed fusion method.

		Input Type	
Algorithm	Artifact	RGB	Other
ADQ1 [23]	JPEG	✓	DCT analysis
DCT [22]	JPEG	✓	DCT analysis
BLK [20]	JPEG	✓	-
CAGI [21]	JPEG	✓	-
Noiseprint [17]	camera-based	✓	-
Comprint [11]	JPEG	✓	-
Comprint + Noiseprint [11]	camera-based & JPEG	✓	-
CAT-Net [27]	JPEG & DCT	✓	DCT filter

**Table 2 jimaging-10-00004-t002:** Datasets used for the evaluation of image forgery detection methods and training of the proposed fusion method.

Dataset	#fake	#real	Format	Notes	Usage
IMD2020 [29]	2010	414	PNG & JPEG	Various forgery types	Training & Evaluation
OpenForensics [30]	18,895	N/A	JPEG	Synthetic face swapping	Training & Evaluation
FaceSwap [31]	879	1651	JPEG	Face swaps with FaceSwap-app	Training & Evaluation
VIPP [32]	62	69	JPEG	Uses double JPEG compression	Evaluation
DSO-1 [33]	100	100	PNG	Only splicing	Evaluation
Coverage [34]	100	100	TIF	Only copy-moves	Evaluation
NC2016 [35]	546	874	JPEG	Splicing, copy-moves and inpainting	Evaluation

**Table 3 jimaging-10-00004-t003:** The average forgery localization performance (F1 score) of every method on every dataset, the theoretical improvement of oracle fusion, the simple fusion methods, and the proposed fusion method. We highlight the three largest values for each column in **bold**, underline and *italics*.

Algorithm	F1 Score						
	VIPP	IMD2020	DSO-1	OpenForensics	FaceSwap	Coverage	NC2016
CAT-Net [27]	0.717	*0.850*	0.675	*0.948*	0.449	*0.573*	0.487
ADQ1 [23]	0.503	0.292	0.420	0.483	0.282	0.211	0.206
BLK [20]	0.431	0.263	0.456	0.263	0.106	0.242	0.233
CAGI [21]	0.437	0.297	0.512	0.294	0.184	0.298	0.293
DCT [22]	0.432	0.313	0.347	0.423	0.191	0.222	0.183
Comprint [11]	0.496	0.297	0.763	0.630	0.351	0.349	0.398
Comprint + Noiseprint [11]	0.581	0.437	0.813	0.711	0.412	0.368	0.439
Noiseprint [17]	0.556	0.396	0.810	0.671	0.347	0.332	0.413
Best oracle fusion of 2 methods	0.807	0.857	0.877	0.961	0.579	0.616	0.594
Oracle fusion of all methods	**0.830**	**0.864**	**0.914**	**0.965**	**0.653**	**0.653**	**0.626**
Maximum	0.700	0.776	*0.822*	0.924	0.435	0.488	*0.514*
Minimum	0.673	0.637	0.793	0.897	*0.558*	0.478	0.476
Proposed fusion method	*0.733*	-	0.753	-	-	0.538	*0.510*

**Table 4 jimaging-10-00004-t004:** The average forgery localization performance (F1 score) of every method on every *test* dataset (i.e., the test split of those datasets), the theoretical improvement of oracle fusion, the simple fusion methods, and the proposed fusion method. We highlight the three largest values for each column in **bold**, underline and *italics*. Note, that the training splits of these datasets were used during training of the proposed fusion method.

Algorithm	F1 Score		
	IMD2020 Test	OpenForensics Test	FaceSwap Test
CAT-Net [27]	0.863	0.952	0.448
ADQ1 [23]	0.340	0.479	0.322
BLK [20]	0.274	0.256	0.134
CAGI [21]	0.316	0.290	0.211
DCT [22]	0.336	0.418	0.220
Comprint [11]	0.424	0.626	0.421
Comprint + Noiseprint [11]	0.457	0.710	0.466
Noiseprint [17]	0.408	0.667	0.409
Best oracle fusion of 2 methods	0.873	*0.961*	*0.584*
Oracle fusion of all methods	**0.883**	0.965	0.647
Maximum	0.806	0.926	0.439
Minimum	0.672	0.898	0.580
Proposed fusion method	*0.869*	**0.966**	**0.814**

**Table 5 jimaging-10-00004-t005:** Evaluation of the detection performance (AUC) of the proposed simple fusion algorithms compared to both the input algorithms and the complete fusion threshold of the input algorithms. We highlight the three largest values for each column in **bold**, underline and *italics*.

Algorithm	AUC					
	VIPP	IMD2020	DSO-1	FaceSwap	Coverage	NC2016
CAT-Net [27]	**0.807**	**0.933**	0.786	0.661	**0.697**	**0.758**
ADQ1 [23]	0.634	0.686	0.654	0.727	0.506	0.603
BLK [20]	0.592	0.553	0.579	0.496	0.508	0.525
CAGI [21]	0.644	0.615	0.650	0.498	0.530	0.558
DCT [22]	0.649	0.611	0.410	0.574	0.505	0.498
Comprint [11]	0.614	0.644	**0.960**	0.526	0.545	0.566
Comprint + Noiseprint [11]	0.630	0.634	**0.960**	0.533	0.556	0.597
Noiseprint [17]	0.608	0.543	0.861	0.517	0.534	0.534
Maximum	0.700	0.803	*0.881*	*0.579*	0.609	0.672
Minimum	*0.734*	*0.750*	0.775	**0.738**	*0.618*	0.696
Proposed fusion method	0.790	-	0.792	-	0.671	*0.686*

## Data Availability

The image forgery detection methods and image forgery datasets used in this proposal are open-source and can be used for research purposes. Links to these methods and datasets can be found in Table 1 and Table 2, respectively.

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
