# Peer review of "Harmonizing Image Forgery Detection & Localization: Fusion of Complementary Approaches"

_2313-433X, 2023, doi:10.3390/jimaging10010004_

Round 1
Reviewer 1 Report
Comments and Suggestions for Authors
The paper proposes a new fusion technique based on a GAN for the forgery localization problem.
First an experimental analysis on existing forensic detectors and their relationship is conducted on state-of-the-art datasets; then, a fusion approach is proposed and tested, which is based on a image-to-image translation framework where multiple heatmaps are transformed into one.
The problem faced is relevant and the proposed approach is interesting.
The experimental analysis shows that for some combinations of training/evaluation datasets, the proposed method brings a significant improvement over individual detectors, and also over the so-called theoretical oracle fusion (optimal but impractical in real-world scenarios). For some other data combinations, the results are still typically higher than individual detectors but way below the theoretical limit, thus stil exposing generalisation challenges.
I have the following comments on the submission:
* I find confusing that results obtained with the proposed fusion method are presented in Table 3 before discussing the method itself, and then later in Table 4 with a different training/evaluation data combination. Also, Table 5 goes back to the previous training/evaluation split. I suggest to discuss all the results together, explaining the different settings
* I understand comparing the proposed method with previously proposed fusion techniques is challenging but simple fusion strategies could be used as reference, such as pixel-wise OR, AND, MAJORITY VOTING rules at a fixed threshold among the different heatmaps.
* I suggest to include in the bibliography (and consider for future extensions) the following forensics detectors:
M. Huh, A. Liu, A. Owens, A. A. Efros, Fighting fake news: Image splice detection via learned self-consistency, in: Proceedings of the European Conference on Computer Vision (ECCV), 2018.
A. Ghosh, Z. Zhong, T. E. Boult, M. Singh, Spliceradar: A learned method for blind image forensics, in: IEEE Conference on Computer Vision and Pattern Recognition Workshops, 2019.
* At line 243, the sentence is incomplete “and that for every image in the dataset."
Reviewer 2 Report
Comments and Suggestions for Authors
A well-timed paper on image tamper detection when deep fake and other image tampering has become a problem. In this paper, a fusion approach that combines multiple individual tamer detection has been proposed. The shown results are an improvement over other existing works. The paper is well written.
Author Response
Thank you for your review. We are happy that you recommend our paper for acceptance without any revisions.
For your information, we have uploaded a revised version according to the other reviewers' comments.
Reviewer 3 Report
Comments and Suggestions for Authors
The paper proposes an interesting idea of combining image forgery detection methods to obtain a better output to help the forensic investigators. Though the idea is quite important but it needs more thorough experimentation. For instance, lack of generalisation on unseen datasets. In forensic problems, it is crucial to deal with mostly unseen stuff. To better consolidate the idea and the contribution, I suggest more efforts should be put towards resolving this issue.
As such paper is well written with a little typo that can be resolved after a thorough proofread of the manuscript. Further, for the comparison with the sate-of-the-art methods incorporate maximum peer reviewed articles not just arrxiv ones.
Comments on the Quality of English LanguageA careful proofread can resolve the issues.
Round 2
Reviewer 3 Report
Comments and Suggestions for Authors
Thank you for your response to my comments. I am convinced with your response.